# Characterization of Slurry-Cast Layer Compounds for 3D Printing of High Strength Casting Cores

**DOI:** 10.3390/ma14206149

**Published:** 2021-10-16

**Authors:** Patricia Erhard, Jan Angenoorth, Joachim Vogt, Johannes Spiegel, Florian Ettemeyer, Wolfram Volk, Daniel Günther

**Affiliations:** 1Fraunhofer Institute for Casting, Composite and Processing Technology IGCV, Lichtenbergstr. 15, 85748 Garching, Germany; johannes.spiegel@igcv.fraunhofer.de (J.S.); florian.ettemeyer@igcv.fraunhofer.de (F.E.); wolfram.volk@utg.de (W.V.); daniel.guenther@igcv.fraunhofer.de (D.G.); 2Chair of Metal Forming and Casting, Technical University of Munich (TUM), Walther-Meissner-Str. 4, 85748 Garching, Germany; jan.angenoorth@tum.de; 3Center for High-Temperature Materials and Design HTL/Fraunhofer Institute for Silicate Research ISC, Gottlieb-Keim-Str. 62, 95448 Bayreuth, Germany; joachim.vogt@isc.fraunhofer.de

**Keywords:** additive manufacturing, casting cores, ceramics, characterization, drying, sintering, slurry, 3D printing

## Abstract

Additive manufacturing of casting cores and molds is state of the art in industrial application today. However, improving the properties of chemically bonded casting cores regarding temperature stability, bending strength, and surface quality is still a major challenge. The process of slurry-based 3D printing allows the fabrication of dense structures and therefore sinterable casting cores. This paper presents a study of the slurry-based fabrication of ceramic layer compounds focusing on the drying process and the achievable properties in slurry-based 3D printing of casting cores. This study aims at contributing to a better understanding of the interrelations between the drying conditions in the 3D printing process and the properties of sintered specimens relating thereto. The drying intensity influenced by an IR heater as well as the drying periods are varied for layer thicknesses of 50, 75, and 100 µm. Within this study, a process window applicable for 3D printing of sinterable casting cores is identified and further indications are given for optimization potentials. At layer heights of 75 µm, bending strengths between ~8 and 11 MPa as well as densities of around 50% of the theoretical density were achieved. Since the mean roughness depth *Rz* is determined to be <30 µm in plane, an application of slurry-based 3D printing in investment casting is conceivable.

## 1. Introduction

ISO/ASTM 52900 describes additive manufacturing (AM) as a process of directly producing shapes from 3D data in which, in contrast to subtractive and formative methodologies, this is achieved layer upon layer. This powder-based 3D printing technique, often also referred to as the “binder jetting process”, was initially developed by Sachs and has been used for printing molds for metal casting from the outset [1]. The low density, stability, and surface quality of the dryly built parts as a result of limited powder flowability for small particle sizes were soon found to clearly restrict the area of application. Almost one decade later, Sachs explicitly described the layer-wise deposition of fine powders dispersed in a liquid in a second patent: a strategy to overcome this constraint [2]. The usage of fine particle suspensions instead of dry powders leads to increased packing densities. In a subsequent sintering step, even full densification can be achieved [3,4].

### 1.1. Slurry-Based 3D Printing in Comparison to Conventional 3D Printing of Sand Cores

In metal casting, removable cores are used to shape the internal cavities of cast components. Fundamental requirements for sand core properties include strength, storage properties, erosion and penetration resistance, chemical resistance to the cast metal, gas permeability, easy decoring, recyclability, and environmental friendliness [5]. The process development of slurry-based 3D printing related to this study aims primarily at the improvement of surface qualities, strengths, and temperature resistances of casting cores compared to those produced by powder-based 3D printing. An enhancement regarding these key features is expected to pave the way to a comprehensive exploitation of the design freedom in additive manufacturing.

Even though the print head characteristics still limit the printing resolution within each layer, slurry-based 3D printing creates the prerequisite for lowering both particle sizes and layer heights and thus for significant improvements in surface qualities. The typical average roughness depth *Rz* for castings produced by lost sand molds varies between 30 and 360 µm, strongly depending on the molding material and process. Elevated surface requirements apply to investment casting (*Rz* between 6 and 30 µm) [6]. For the herein-investigated slurry-based 3D printing process, we achieve surface qualities in the range of those in plane, while in building direction, the selected layer height will be decisive.

Relevant binder systems in 3D printing of sand cores include furanic and phenolic resins as well as inorganic binders. Phenolic resins typically show the highest temperature stability among the organic alternatives. Inorganic binder systems exhibit an improved temperature stability since no decomposition takes place [7]. However, parts produced thereof typically show plastic deformation at high temperatures [8]. The temperature stability of the binder system itself is of minor importance with casting cores manufactured by slurry-based 3D printing since the binder is only used for depicting the geometry and burnout during sintering. In contrast, the material composition of the slurry and the sintering process parameters strongly affect the resulting mechanical and thermal properties in slurry-based 3D printing [9].

Bending strength is considered to be the crucial mechanical property for sand cores in casting applications [5]. Typically, 3D-printed casting molds and cores achieve bending strengths between 2.5 and 5 MPa [10]. Previous investigations of similar types of slurry showed an adjustability of the 3-point bending strength by slurry additives and sintering parameters up to 18 MPa [9]. Since the lost structures must be removable after casting, casting cores are intended to exhibit strengths sufficient to withstand the loads during handling and casting while assuring destructibility after casting [11,12].

### 1.2. Literature on Slurry-Based 3D Printing Technology

The authors refer to Zocca et al. [3] and Travitzky et al. [13] for an overview of additive manufacturing processes for ceramics in general, comparing available technologies and materials. The slurry-based formation of layers is considered to be a process combining characteristics of two conventional processes: tape casting and slip casting. Like in tape casting, the slurry is spread precisely in a thin layer and, similar to slip casting, a wet layer is bound to the previous porous dry layer by adhesion [14]. Subsequently, the focus is set on the slurry-based formation of layers, which has been studied previously as a basis for both 3D printing and selective laser sintering using varying dispersion media, particle sizes, and materials. Table 1 gives an overview of studies conducted in slurry-based additive manufacturing technologies without claiming to be exhaustive. In contrast to other known research papers, Yen [14] is working with a PVA solution instead of an aqueous dispersion and demonstrates a slurry-feeding system delivering slurry on demand during the coating process. Further applications are also known using materials such as glass and glass ceramics [15]. Vogt et al. [9] describe the development of a slurry material and the respective sintering process explicitly, focusing on the requirements for casting applications. This work builds upon recognition thereof.

### 1.3. Aims of This Study

The scope of this article is to show the impact of the process parameters related to drying on the properties of layer compounds formed by repetitively applying slurry on a building platform. Stress development during drying that causes cracking and deformation has been studied for decades and was found to result from a pressure gradient within a porous drying body [19]. Scherer [19] reported an increase in stress with increasing drying rates and body sizes and a decrease in stress with increasing permeability. Thus, it is expected that the drying parameters will affect the physical properties of each single layer and the layer compounds.

The test setup selected for this study (Section 2.2) allows varying the drying conditions based on the parameters of the IR heater intensity and drying period. Furthermore, the layer heights are altered between 50 and 100 µm. Thereby, the intention is to recognize interrelations regarding size dependency. The density, bending strength, and microstructure of subsequently sintered layer compounds as well as the surface roughness in the green state are analyzed. The characterization of layer compounds points out potentially applicable parameter sets and contributes to a deeper understanding of the impact of drying conditions on the properties of slurry-based 3D printed parts.

## 2. Materials and Methods

### 2.1. Materials

The quartz-water-based slurry Q24 used is a variant of the slurries investigated in [9]. It consists of 44 vol.% quartz flour (Sikron SF500, Quarzwerke GmbH, Frechen, Germany; d_50_ value 4 µm, d_95_ value 13 µm), about 55 vol.% of deionized water, and 1 vol.% organics, consisting in turn of a dispersing agent, a viscosity modulating agent, a defoamer, and a preservative. The slurry exhibits excellent stability and favorable flowing behavior while showing no agglomeration and foaming.

### 2.2. Test Setup

Figure 1 shows a scheme of the key items for the experimental study conducted herein as parts of a fully automated slurry-based 3D printing test setup. The investigated process of fabricating ceramic layer compounds is comprised of the following steps: firstly, the gap between the coater and the building platform is adjusted to the required layer height. Subsequently, a slurry coater applies slurry to the building platform whilst moving. The feeding rate corresponds to the coater velocity and the predefined layer height. Slurry is thus supplied on demand with little excess material. The step of coating is followed by the step of drying. Therefore, an IR heater is centered above the building platform and heats the layer until it emits the required amount of thermal radiation, which is monitored by a pyrometer. The building platform is lowered before the coater moves back to the start position. This step normally also comprises the spraying of a binder onto the dried layer. In this study, no binder is applied to analyze the layer compounds while avoiding superimposing effects.

The drying setup includes two measuring points: The first one is used to acquire the radiation data of the top of each layer via pyrometer and to actively control the drying process. The second one depicts the tip of a thermocouple and serves as a reference for aligning the pyrometer data and the measured temperatures. This setup leads to declining drying periods for individual layers within the production of a single specimen.

The experimental test setup is shown in Figure 2 and comprises two moveable axes driven by stepper motors, one for the coating module and one for the building platform. A programmable logic controller (PLC) based on SIMATIC S7 (Siemens AG, Munich, Germany) is used to control the test setup, including the motion of the axes, a directional control valve, and the IR heater.

The coating module includes the pneumatically actuated slurry feeding system. The IR heater (IRD S230, Optron GmbH, Garbsen, Germany) is mounted on the coater module at a fixed position, which ensures homogeneous heating of the specimen during drying. An appropriate position of the IR lamp was determined by examining the temperature homogeneity within the building area using a thermal imaging camera PI400i (Optris GmbH, Berlin, Germany). The distance from the respective layer to the filament of the IR-lamp was adjusted to 65 mm. To prevent residual slurry drying and clogging the coater during the drying process of each layer, a cleaning unit consisting of a constantly rotating roller dipped in water is positioned in such a way that the coater rests upon the cleaning unit while each layer is drying.

A pyrometer (CT-SF22, Micro-Epsilon Messtechnik GmbH & Co. KG, Ortenburg, Germany) is installed above the building platform and captures the emitted energy of a freshly applied ceramic layer during drying, as depicted in Figure 1. The radiation of the layer itself is superimposed by the infrared radiation of the IR heater. Furthermore, the emission coefficient of the layer is expected to change over time due to moisture loss. Therefore, the temperature values displayed by the pyrometer are subsequently treated as digital values representing a certain state of drying. A type K thermocouple is connected to a universal measuring amplifier (QuantumX MX840B, Hottinger Brüel & Kjaer GmbH, Darmstadt, Germany) to monitor the real temperature occurring in the undermost layer.

### 2.3. Fabrication of the Layered Structures

#### 2.3.1. Experimental Plan

An acrylic glass plate, roughened by sandblasting, was clamped onto the building platform and serves as substrate for each experiment. The coating velocity was set to a constant value of 34 mm/s throughout the entire experimentation. Table 2 shows the varied parameters within this study. In total, 27 experiments were performed, resulting in layer compounds of approximately 30 × 60 × 3 mm^3^ each. Experiments for the layer heights of 50, 75 and 100 µm were conducted for different drying periods (hereafter referred to as drying points) and drying intensities (corresponding to the power adjustment voltage (PAV) of the IR heater). Since different layer heights were preset and the coater applies slurry on demand with little excess material, the material flow at the coater outlet was adjusted to the respectively demanded amount beforehand. The finished samples were stored in a drying cabinet at a relative humidity of 10% and temperature of 20 °C.

#### 2.3.2. Definition of the Drying Points

The meaning of the drying points will be discussed shortly in the context of the theoretical background. From drying technology, it is known that as soon as the air above a specimen is saturated with humidity and the temperatures of the air and the surface of a product to be dried are nearly equal, a state of equilibrium called the adiabatic saturation temperature is reached [20]. Drying processes (Figure 3) start with the first stage of drying, also called the constant rate period, where the pores are completely filled with water [19]. In the subsequent first falling rate period, air enters the pores while the majority of evaporation still occurs at the exterior surface [19]. Evaporation drives the meniscus of the liquid further into the interior, causing the second falling rate period. This stage is associated with evaporation inside the drying product and a convergence of the surface temperature with the ambient temperature [19]. For further details on drying technology and drying kinetics of ceramic suspensions, the authors refer to [19,21]. The stages of drying are considered to be well-represented by the near-surface measurement of the pyrometer and are therefore included in the selection of the drying points.

To define the drying points in the process, the control unit monitors the pyrometer signal and aborts the drying step as soon as the previously determined setup value is obtained. The emissivity ε of the pyrometer is set to 1. Based on experiments, the Drying Point 3 that has very low amounts of residual moisture within each layer is selected for the three levels of IR heater PAV. Temperature data are acquired for these at layer heights of 100 µm.

Figure 4 shows the determination of the Drying Points 1 and 2 based on these. An increase in temperature is determined in the early stage by both the pyrometer and thermocouple measurements. Drying Point 1 represents the digital value of the pyrometer at the local maximum temperature measured by the thermocouple that terminates the early stage. The phase before approximately reaching Drying Point 1 is assumed to represent the first stage of drying (Figure 3a). In the next section, the temperature profile of the pyrometer stagnates, whereas the temperature measured by the thermocouple declines. Drying Point 2 is determined by the local minimum of the thermocouple temperature and represents a digital value of the pyrometer following a longer horizontal course. The authors assume the most significant proportion of evaporation happens between Drying Points 1 and 2, since evaporation causes cooling that corresponds to the first falling rate period (Figure 3b). During the last section, an increase in temperatures is detected again. Between Drying Points 2 and 3, the surface temperature is expected to approach the simultaneously ascending near-surface temperature of the air. This interval is therefore assigned to represent the second falling rate period (Figure 3c), accompanied by evaporation inside the drying product.

Analogous to Figure 4, which corresponds to PAV = 3.4 V, the Drying Points 1 and 2 are, respectively, determined for PAV = 3.8 V and PAV = 4.2 V (Appendix A and Appendix A).

#### 2.3.3. Cutting of Specimens

Before sintering, the samples are cut into cuboids using the Brillant 220 cut-off machine (ATM QNESS GmbH, Mammelzen, Germany) and a diamond cut-off wheel.

The specimens are cut perpendicular to the direction the samples were deposited and carefully scraped off the substrate. The width of the specimens is intended to comply well with DIN 843-1:2008-08, embodiment A.

#### 2.3.4. Sintering

In preceding work [9], the sintering behavior of slip cast specimens was analyzed by using thermo-optical measurement devices and conducting sintering tests. It was shown that by sintering at 1200 °C, minor shrinkage occurs, yielding a slightly increased density and comparatively low bending strengths of about 4.5 MPa. When sintered at 1300 °C for 5 h, a minor sample inflation of up to 0.7% occurs, which was caused by cristobalite formation. Despite the slight decrease in density, the three-point-bending strength reached about 11 MPa. Further results indicated that sintering slightly below 1300 °C yields comparative bending strength values while showing less sample inflation. Therefore, the target sintering temperature is chosen to be 1275 °C with a dwelling time of 5 h. The specimens are sintered in the Top 60 furnace model (Nabertherm GmbH, Lilienthal, Germany). An analysis of the sintering behavior is not included in this article. However, building upon the results of [9], relevant deformation due to sintering shrinkage is unlikely to occur.

#### 2.3.5. Surface Finish for Bending Strength and Density Determination

The surface finish was intended to meet the requirements of DIN EN 843-1:2008-08. Therefore, the cut and sintered specimens are ground to the required dimensions with grade 400 grit-paper and a standard caliper gauge and finished with grade 2000 grit-paper. To ensure perpendicularity, a cuboid made of steel is used as a grinding jig for manual grinding. The sharp edges are carefully beveled.

### 2.4. Examination Methods

#### 2.4.1. Density

The length of the sintered cuboids is measured using a standard caliper gauge, and the cross section is measured with a micrometer. The weight m is determined by a Secura 15-1S analytical balance (Sartorius AG, Göttingen, Germany). The density ρ follows:

ρ = m/V
(1)

#### 2.4.2. Four-Point Bending Strength

The four-point bending strength is determined based on the standard DIN EN 843-1:2008-08 using the Inspekt Table 100 kN universal testing machine (Hegewald & Peschke Meß- und Prüftechnik GmbH, Nossen, Germany) and a 5 kN load cell.

#### 2.4.3. Surface Roughness

The surface roughness of the 27 samples is measured perpendicular to the coating direction prior to sintering using the MarSurf M400 surface measuring instrument (Mahr GmbH, Göttingen, Germany) and the BFW A 10-45-2/90° probe arm.

#### 2.4.4. Microscopy

Micrographs are captured using the BX53M microscope (Olympus Europa SE & Co. KG, Hamburg, Germany).

## 3. Results

The following section describes the results. For easy readability, the test series are named after the corresponding parameters for each experiment (Table 2), omitting the respective units starting with the layer height, followed by the drying point, and the IR heater PAV, e.g., 100;1;3.4 for the parameter set h = 100 µm; Drying Point 1; IR heater PAV = 3.4 V.

### 3.1. Analysis of In-Process Temperature Curves

A uniform distribution of heat within the sample is crucial for stress-free drying. Overall, ensuring minimal variations in temperature within the process is considered important to most additive manufacturing processes and affects the properties of the resulting parts. For the slurry-based 3D printing technology, the impact of temperature and moisture within the layer compound is assumed to be correlating and superimposing simultaneously.

Figure 5 shows a typical temperature profile recorded within the process. The pyrometer signal actively impinges on the process, in particular the drying periods, by delivering digital values of the controlled variable that is queried during the drying step of each individual layer. In contrast, the thermocouple cast into the undermost layer monitors the actual temperature profile within the layer compound.

Figure 6 outlines the temperatures measured by thermocouple at the respective Layer 5 (corresponding to “Start”), compared to those acquired at the bottom layer when applying the last layer to each sample for different experiments. At higher drying intensities, increased absolute temperatures are determined. Additionally, longer drying periods result in higher temperatures. The attained absolute temperatures increase with lower layer heights, following the same trend for each layer height. While the temperatures at the uppermost layers differ between 53.5 and 78.4 °C at a layer height of 50 µm, they vary between 50.3 and 65.8 °C for the layer height of 100 µm and between 51.0 and 71.9 °C for the layer height of 75 µm. The decreasing absolute temperature for thicker layers is assigned to an increased amount of water being applied, which may result in a higher volume of liquid not being evaporated directly but soaked up by the underlying layers and cooling the sample. The test series with PAV = 3.4 V and Drying Point 3 are identified as outliers in Figure 6, showing noticeably higher absolute temperatures. This is contributed to an experience-based selection of the Drying Points 3. At low drying intensities, the material is expected to be more tolerant regarding excessive drying since stresses are reduced.

All test series result in specimens, indicating that an appropriate process window for further investigations has been found, using the parameter sets as depicted in Table 2. The method of defining the drying points, as described in Section 2, is thus considered to be suitable. Nevertheless, there is potential for optimization regarding an increased uniformity of absolute temperatures during the layering process, which is assumed to further improve the overall quality of built parts.

### 3.2. Physical Properties

To analyze the physical properties, it is crucial to be aware of an inhomogeneity within each layer compound corresponding to one experiment. Regarding the process of applying slurry on demand, the start-up area is critical for defects. Eight specimens extracted from the middle to the rear part of each sample are used to analyze the achieved densities. Subsequently, five respective specimens of each experiment are analyzed in four-point bending tests. The number of samples n is noted in the diagrams’ subtitles. The results are checked for significance using pairwise multiple comparisons [22].

The specimens for Drying Point 3 are found to be highly fragile. Therefore, only a few specimens are free of major defects prior to sintering. The respective experiments are excluded in the following subchapters analyzing density and bending strength. Instead, a microscopic analysis will provide a closer look at the microstructure.

All specimens are sintered and ground to the required dimensions prior to analysis.

#### 3.2.1. Density

Analysis of the densities of the sintered specimens allows the prediction of beneficial parameter sets in slurry-based 3D printing. Density is considered to be a crucial property in forming processes of ceramics. High green densities are known to positively affect the sintered density as well as the shrinkage ratio [23]. However, high densities are accompanied by reduced permeability. Drying bodies of reduced permeability, in turn, are expected to be more vulnerable to stresses during drying [19] and might thus be critical for the overall process.

Herein, the density in the sintered stage is investigated since the particles of the layer compounds are not bound by a binding agent but only agglomerated during drying. The specimens thus gain sufficient strength for handling by sintering only.

Figure 7 shows the density distributions for varied values of IR PAV and indicates higher densities at Drying Point 2 compared to Drying Point 1. For most of the test series, lower layer thicknesses positively impact the density of the specimen. However, one remarkable deviating test series (50;2;3.4) stands out. The cause of this unexpected behavior is unknown and might be related to an untraceable deviation in the fabrication process since an in-process temperature profile strongly differing from the ordinary course, depicted in Figure 5, was found. Further investigations are required to identify whether the behavior is systematic.

Additionally, Figure 8 shows that higher densities are achieved at low PAVs of the IR heater. At h = 100 µm, where the lowest densities occur in general, no trend can be identified. However, the results from h = 50 µm and h = 75 µm clearly suggest low heating intensities and therefore slow drying rates being beneficial for achieving high densities. The reason might be a reduced level of stress for a slow and therefore increasingly homogenous drying, as well as an increased amount of water that is soaked into the layers underneath instead of evaporating immediately on the surface, leading to an increased green packing density caused by particle rearrangement and capillary forces.

A statistical two-sample location test [22] is utilized to confirm the hypotheses stated above and to assess the statistical significance of the results. The significance level α is set to 0.05.

A total of 62.5% of the mean density values acquired for layer height h = 100 µm are found to differ significantly from those for the lower layer heights. Furthermore, 47.6% of the mean densities for Drying Point 1 can be significantly distinguished from those for Drying Point 2. A total of 45.8% of the mean densities measured for PAV = 3.4 V are significantly distinguishable from the higher IR heater intensities. The statistics indicate an overlap of data for the test series. However, the overall average percentage of significance between all samples equals 41.1%. This supports the hypothesis of a causal relationship as stated above.

#### 3.2.2. Four-Point Bending Strength

Analyzing the strength of brittle materials is a highly challenging task, since statistically distributed flaws usually initiate fractures within brittle material. Strength is dependent on the size of the major flaw as well as the size of specimen, since flaws are more likely to appear in large specimens. Therefore, the strength of ceramics must be described with statistics [24]. A detailed and proofed statistic on bending strength distributions cannot be provided in this study, since it demands an increased number of samples for every test series. However, the bending strength distributions will give a first guidance on process-related contributing factors.

Figure 9 depicts the bending strengths for different drying intensities. While (a) (PAV = 3.4 V) shows rather low strengths at low layer heights, (c) indicates the opposite effect with PAV = 4.2 V. This effect may result from an enhanced compatibility of low layer heights with high drying intensities and of high layer heights with low drying intensities, regarding stress-free drying.

Neglecting the series with layer heights of 50 µm, longer drying periods (Drying Point 2) seems to positively affect the specimens when using higher intensities, whereas again the opposite is suggested at PAV = 3.4 V.

In Figure 10, the respective bending strengths are depicted regarding the layer heights at Drying Point 1. Figure 10 indicates that lower drying intensities might be beneficial for higher layer thicknesses (a) and disadvantageous for lower layer thicknesses (c). An increased formation of stresses at higher drying rates for large layer heights than for thin films and the high absolute volume of water being rapidly evaporated might be the reason for a creation of major defects.

The statistical test shows only two test series differing significantly at α = 0.05. The series 75;2;3.8 differs significantly from 50;1;4.2 as well as from 100;1;3.4. Therefore, no statistically relevant conclusions can be derived for the bending strength results. However, the comparison gives only hints on process parameters tending to provoke flaws in the used setup. A higher number of specimens is necessary to verify the results. Regardless of the PAV setting, four-point bending strengths of ~10 MPa were achieved. The magnitude of bending strengths complies with the results of three-point bending tests while developing the sintering process [9]. In further investigations and industrial application, an adjustment of the physical properties in the sintering process is proposed to achieve the respective desired properties. At the same time, an optimization of the 3D printing process regarding an economic build-up rate is suggested. High drying intensities result in high build-up rates and are therefore favorable in economic terms.

#### 3.2.3. Microscopy

Previous investigations showed that the layer height reaches the preset value at Layers 10–15 for layer heights of 50 to 100 µm. Herein, no samples are analyzed below Layer 15. As mentioned above, the samples of Drying Point 3 did not show sufficient handling strength to be analyzed for density and bending strength.

Figure 11 presents a comparison of two specimens of the same layer height and drying intensity, but different drying points. Major flaws are visible for Drying Point 3 in (b) compared to Drying Point 1 (a). Nevertheless, (b) exhibits a smaller proportion of porosity, indicating an overall increased density.

Figure 12 allows a closer look at those areas. The images of the samples depicted in (a), (b), and (c) show approximately similar proportions (~30% dense, darker areas/~70% porous, brighter areas) of two phases representing one layer. In contrast, a highly dense structure pervaded by a thin layer of accumulated pores can be observed in (d). From fracture mechanics, it is known that notches smaller than the intrinsic defects (looking at porous ceramics represented by the pores) are not decisive. Therefore, a critical size of non-damaging notches for porous materials is discussed in the literature [25]. The flaws and accumulated pores detected in Figure 11b and Figure 12d are suspected of being critical in conjunction with the overall relatively dense structure. Further investigation focusing on these observations and the respective results of density and bending strength is suggested.

The layers of increased density are assigned to the near-surface area. The direct contact with radiation and convective heat is assumed to cause excessive drying within this area, resulting in local compression during shrinkage of each layer. The specimen shown in Figure 12d was produced at absolute temperatures of ~60–80 °C (measured at the undermost layer). A more intense densification is apparent in approximately 90% of the layer. Dependent on requirements in application, it might be a conceivable solution to configure the microstructure as needed in the respective casting core or even region of the casting core.

### 3.3. Surface Roughness

The surface roughness of the specimens is analyzed on top of the uppermost layers and perpendicular to the coating direction for further hints on beneficial parameters for slurry-based 3D printing. Figure 13 shows the average roughness depths *Rz* for the investigated layer heights and points out a constant level of roughness for layer heights of 100 and 75 µm, while increased roughness values become visible at h = 50 µm. However, no significant differences between the test series are determined using the multiple comparison procedure. The increased roughness values within this study are assumed to indicate an inhomogeneous coating process at layer heights of 50 µm, since repetitive flaws are considered to cause major irregularities on the surface.

## 4. Discussion

Drying periods between Drying Points 1 and 2 are found to be suitable for slurry-based 3D printing at different parameter sets in this study. Excessive drying until Drying Point 3, which is assigned to the second falling rate period of drying and therefore causes evaporation inside the porous body, has been found to be unfavorable regarding the mechanical properties since major flaws are likely to occur. In porous materials, notches smaller than the intrinsic defects are considered to be non-damaging [25]. Regular porosities within the specimens are considered to be intrinsic defects. The microstructure of the specimens with Drying Point 3 indicates an overall increased density, corresponding to the respective top areas of each layer for other specimens. The densified microstructure is assumed to result in reduced permeability. Moreover, rapid evaporation is expected to occur while making contact with a lower layer that was heated excessively to Drying Point 3. Therefore, an increase in drying stresses due to high drying rates and low permeabilities, as stated in [19], is expected to result in cracks and residual stresses. Assuming material behavior in accordance with the fundamentals in fracture mechanics stated in [25], the mentioned effects of densification and flaw creation may both contribute to the vulnerability of the Drying Point 3 specimens. Further investigations may focus on material and process parameters provoking and preventing flaws to realize highly densified structures without major defects that might be beneficial for certain applications. Approaches might include developing processible slurries of enhanced solid fractions or modifying the drying method.

In general, micrographs showed a densification at the respective near-surface areas of each layer and porosities at the contact with the layer below. Since the near-surface areas are directly exposed to radiation, densification caused by shrinkage is suggested to be a reasonable explanation. Zocca et al. [3] state that due to the free settling of particles, no interface between the individual layers becomes obvious in slurry-based 3D printing. In this study, an interface between the layers was detected for every specimen. Densification of the layer in direct contact with the soaking previous layer, due to rearrangement of particles and capillary forces, was not detected. Future investigations involving binder application are needed to provide evidence that no particles from the freshly applied layer are soaked deeper into the top of the previously cast layer by capillary forces using the slurry material and setup described herein.

Evaluating the density distributions, specimens of lower layer thicknesses exhibit increased densities. Moreover, higher densities are determined at Drying Point 2 compared to Drying Point 1. These findings coincide with the previously stated theory of densification at near-surface areas. However, deeper analysis is required for more evidence.

Overall, layer thicknesses below 100 µm show enhanced properties. Low IR radiation is recommended regarding density, while specimens at h = 50 µm exhibit improved bending strengths at a higher IR intensity. Since flaws and their sizes are decisive in fracture mechanics of brittle materials [25], it is assumed that these are more likely to occur for the test series, resulting in low bending strength. Therefore, specimens containing flaws caused by irregular coating or internal stresses are expected to show low bending strengths, especially at higher densities. However, comprehensive statistics are needed for a reliable statement on favorable parameter sets.

The surface qualities are expected to meet the requirements for investment casting since all gathered data show an average roughness depth *Rz* of <30 µm in plane. The test series at h = 50 µm exhibit comparatively poor surface qualities that are attributed to irregularities caused by the setup for coating on demand. As a consequence, adjustments to the process sequence will be made in order to further optimize the process.

Overall, it can be stated that a process window applicable for 3D printing of sinterable casting cores has been identified. Bending strengths between ~8 and 11 MPa as well as densities between ~1.25 and 1.35 g/cm³, corresponding to theoretical densities around 50%, have been achieved at a layer height of 75 µm. Furthermore, these specimens showed a mean roughness depth *Rz* of 17.5 µm and are thus generally suitable for applications in investment casting.

In contrast to conventional 3D printing material systems for casting applications, binder-related effects such as adhesion to the powder or storage stability are of less importance and not studied herein. Since the binder medium is used only to outline the geometry and constitute a sufficient green strength for handling, the final strength is adjustable by appropriate sintering parameters. However, further research activities involving the whole process chain will lead to more comprehensive process knowledge.

## 5. Conclusions

This article comprises a study of slurry-based fabrication of ceramic layer compounds. A material system suitable for applications in casting technology was used [9]. The study focused on the characterization of specimens fabricated at different parameter sets related to drying, by altering the drying periods and intensities as well as the layer heights. A process window for fabricating sinterable casting cores conceivable for application in the casting industry has been identified. Bending strengths of ~8 and 11 MPa, a theoretical density of ~50%, and mean roughness depths of ~17.5 µm were achieved with layer heights of 75 µm. The process parameters investigated are shown to be beneficial for layer thicknesses lower than 100 µm. The results also indicated that higher densities are be achieved when drying until the first falling rate period. However, further extensive investigations are suggested to provide statistical evidence for the new findings and clear correlations between the process parameters and the physical properties of slurry–cast layer compounds.

In addition, a new method for process control for slurry-based additive manufacturing technologies focusing on drying parameters has been presented and its suitability has been shown.

Specified requirements for the application in casting technology will influence the further focus on developments, since both porous and highly densified casting cores may be beneficial depending on the respective application and decoring process.

Overall, slurry-based 3D printing of sinterable casting cores shows high potential for casting applications. Temperature-stable high-strength casting cores are achievable at surface qualities approaching those required for investment casting applications while using well-established and scalable binder jetting systems.

## Figures and Tables

**Figure 1 materials-14-06149-f001:**
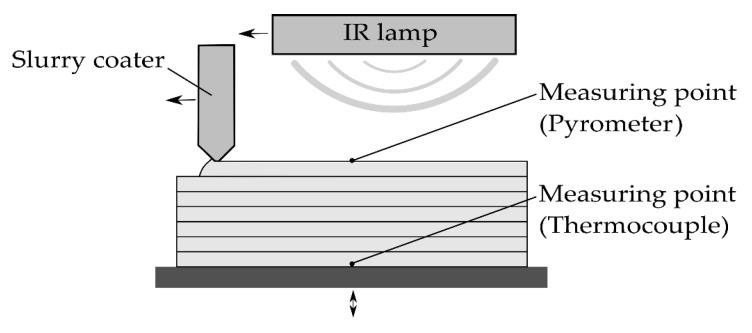
Scheme of the investigated process.

**Figure 2 materials-14-06149-f002:**
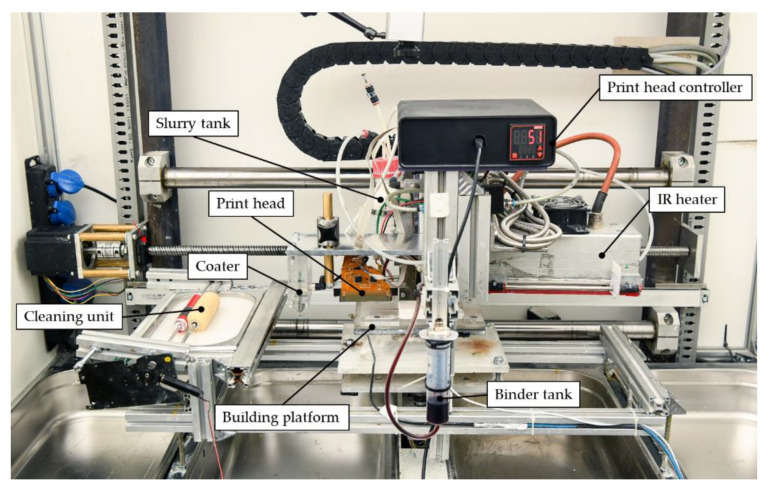
Slurry-based 3D printing test setup.

**Figure 3 materials-14-06149-f003:**
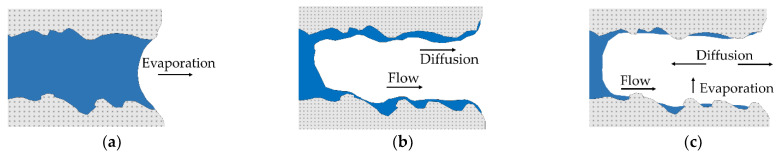
Scheme of the theory of drying based on [19] associated with the experimentally determined drying points. (**a**) constant rate period (Drying Point 1); (**b**) first falling rate period (Drying Point 2); (**c**) second falling rate period (Drying Point 3).

**Figure 4 materials-14-06149-f004:**
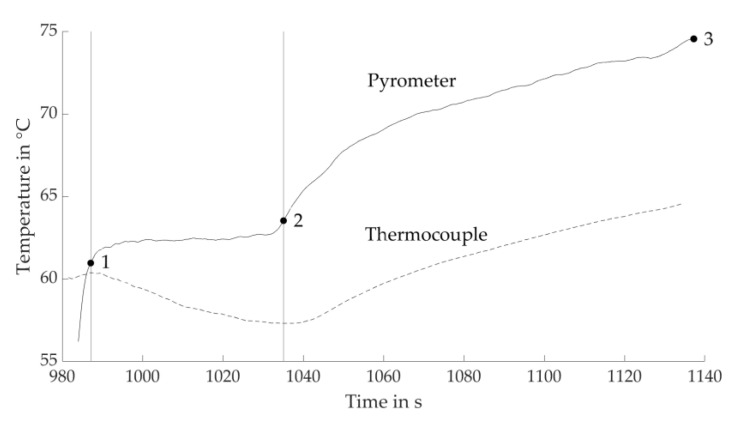
Determination of Drying Points 1 and 2 by comparing the temperatures measured by thermocouple and the pyrometer signal at an emissivity of 1 (digital values) and Layer 5. The signals are recorded in the parameter sets with Drying Point 3 corresponding to the estimated boundary of the process window. Drying Point 1 corresponds to the local maximum measured by thermocouple, Drying Point 2 to the local minimum. Exemplary for PAV = 3.4 V at a layer height of 100 µm.

**Figure 5 materials-14-06149-f005:**
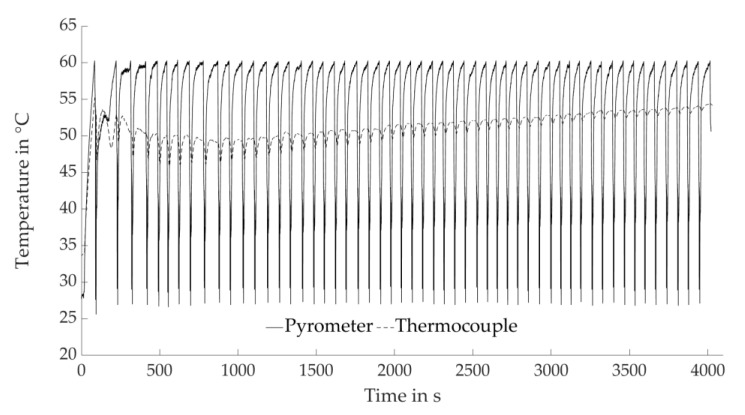
Exemplary in-process temperature profile (50;1;3.4). Comparison of the pyrometer signal at an emissivity of 1 (digital values) and the temperatures measured by thermocouple.

**Figure 6 materials-14-06149-f006:**
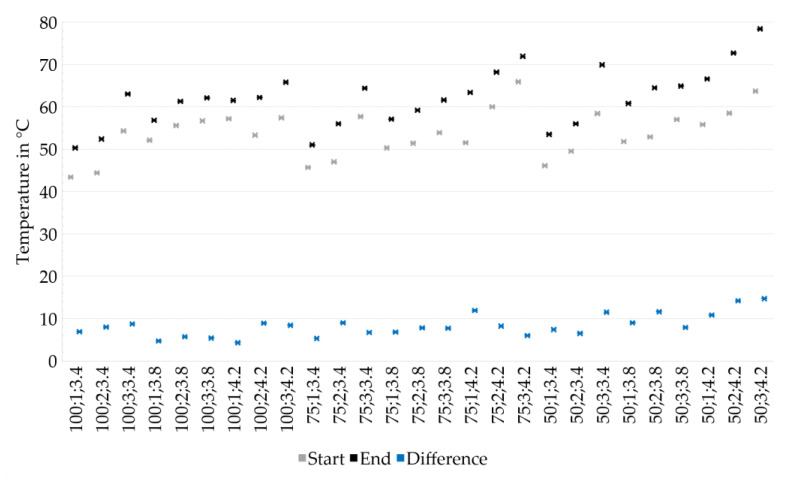
Analysis of temperature increases within the process, measured by the thermocouple. Temperatures increase with decreasing layer heights and increasing drying intensities and drying periods.

**Figure 7 materials-14-06149-f007:**
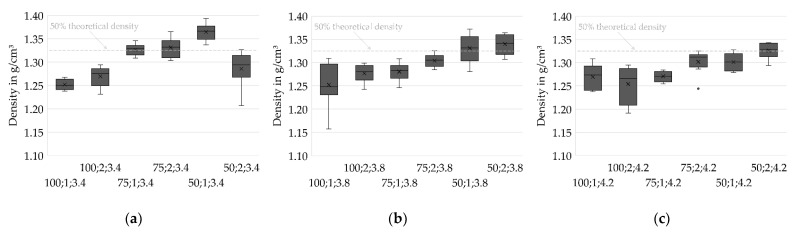
Density distributions for different drying intensities. A reduction in layer thickness results in higher densities. Drying Point 2 shows higher densities in most cases. (**a**) PAV = 3.4 V, *n* = 8; (**b**) PAV = 3.8 V, *n* = 8; (**c**) PAV = 4.2 V, *n* = 8.

**Figure 8 materials-14-06149-f008:**
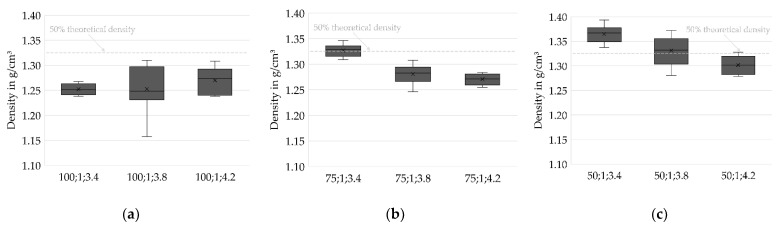
Density distributions for different layer heights. Higher drying intensities result in reduced densities for h = 75 µm and h = 50 µm. For h = 100 µm, no trend can be deduced. (**a**) h = 100 µm, *n* = 8; (**b**) h = 75 µm, *n* = 8; (**c**) h = 50 µm, *n* = 8.

**Figure 9 materials-14-06149-f009:**
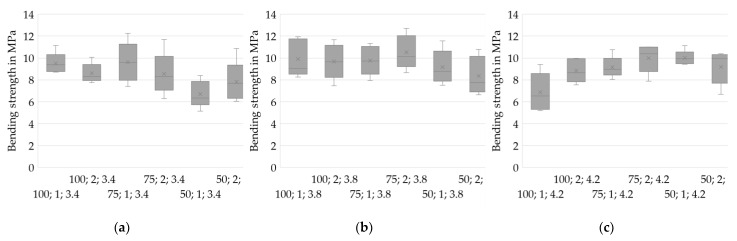
Four-point bending strengths for various drying intensities. Contrary indications: while (**a**) indicates higher strengths at higher layer heights, (**c**) displays tendencies to higher strengths at low layer heights. (**a**) PAV = 3.4 V, *n* = 5; (**b**) PAV = 3.8 V, *n* = 5; (**c**) PAV = 4.2 V, *n* = 5.

**Figure 10 materials-14-06149-f010:**
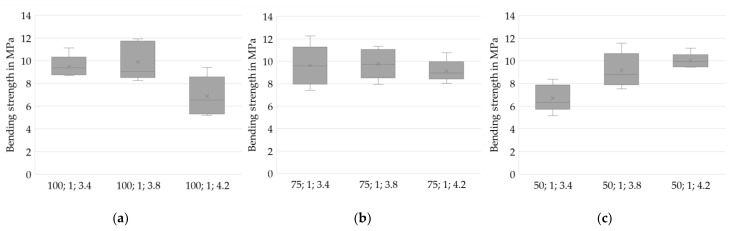
Four-point bending strengths for various layer heights. Higher drying intensities result in reduced strengths for h = 100 µm (**a**) and in increased strengths for h = 50 µm (**c**). For h = 75 µm, no trend can be deduced (**b**). (**a**) h = 100 µm, *n* = 5; (**b**) h = 75 µm, *n* = 5; (**c**) h = 50 µm, *n* = 5.

**Figure 11 materials-14-06149-f011:**
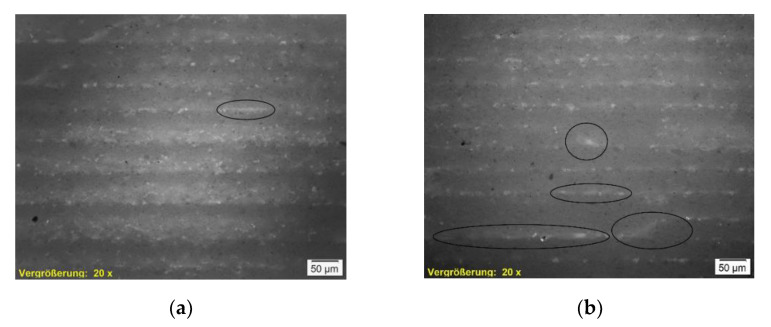
Microscopic images of selected samples (magnification 20×) depicting major flaws for Drying Point 3 (**b**) compared to Drying Point 1 (**a**). Nevertheless, (**b**) shows smaller proportions of porosities, indicating increased density. (**a**) 50;1;4.2; (**b**) 50;3;4.2.

**Figure 12 materials-14-06149-f012:**
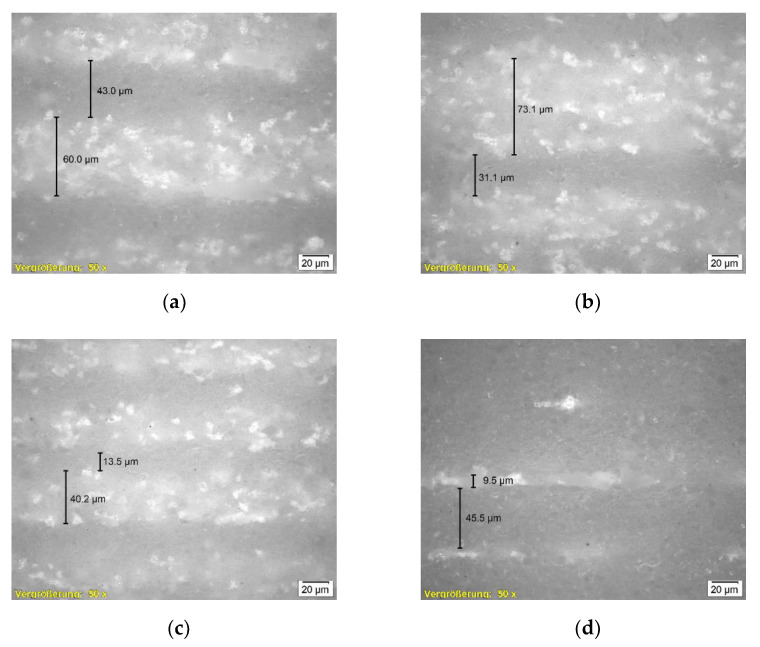
Microscopic images of selected samples (magnification 50×) depicting a microstructure split into two layers at similar proportions for (**a**–**c**). The visible structure in (**d**) shows a remarkable dense structure together with small areas of accumulated pores. (**a**) 100;1;3.4; (**b**) 100;3;4.2; (**c**) 50;1;4.2; (**d**) 50;3;4.2.

**Figure 13 materials-14-06149-f013:**
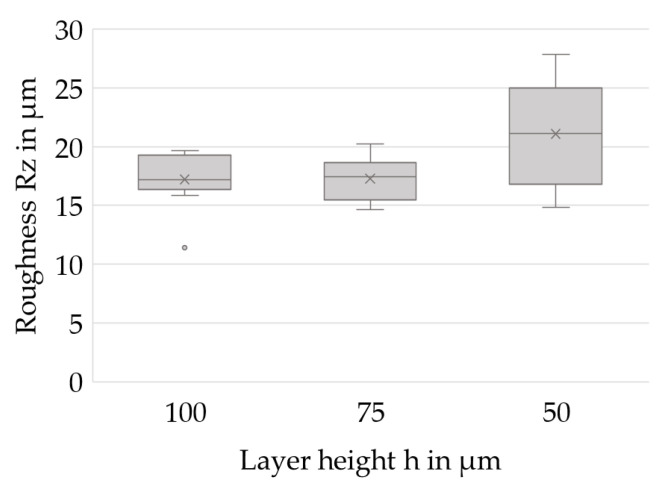
Average roughness depths *Rz* for different layer heights.

**Table 1 materials-14-06149-t001:** Overview of Exemplary Studies Conducted within Slurry-Based Additive Manufacturing.

Material	Dispersion	Solid Load	Average Particle Size in µm	Layer Thickness in µm	Theoretical Density/Shrinkage Ratio in %
Al_2_O_3_-SiO_2_ [16]	Aqueous	63–87 wt.%	N/A	100	<90/<10
Porcelain [17]	Aqueous	87 wt.%	8.3	100	<87/<10
Porcelain [18]	Aqueous	N/A	N/A	100–200	N/A/<18
Al_2_O_3_ [4]	Aqueous	34 vol.%	0.5	50	99.7/<19.2
SiO_2_ [14]	Polyvinyl alcohol	46.84 wt.%	<10	10–80	N/A/N/A

**Table 2 materials-14-06149-t002:** Varied Parameters in the Coating Experiments.

Layer Height in µm	Drying Point	Power Adjustment Voltage PAV of IR Heater in V
50, 75, 100	1, 2, 3	3.4, 3.8, 4.2

## Data Availability

The data presented are available on request from the corresponding author. The data are not publicly available since they are part of an ongoing study.

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
