# Peer review of "Characterization of Slurry-Cast Layer Compounds for 3D Printing of High Strength Casting Cores"

_materials, 2021, doi:10.3390/ma14206149_

Round 1
Reviewer 1 Report
I think this paper is quite interesting experimental results but this paper are written in a way that is quite difficult to understand. It is necessary to write the condition of each experimental result in detail.
- The temperature was measured only at the center point, but if the IR lamp is not too large or too far from the printing area, the temperature is likely to be different depending on the location. Did you check that there is no temperature deviation depending on the drying location?
- It would be better to show the description of the drying point in lines 192~203 as a schematic diagram. In addition, it would be good to present a graph of how the drying point was set in other experimental conditions as a supplementary material. It is also necessary to show the drying point as time for each experimental condition.
- Section 2.4.1: Unless there is a premise that the dimensions are uniform, the density must be measured in another way. In particular, since used specimen could be deformed during drying or sintering process, the precision of the proposed density measurement method is questionable. If the specimen is in a state after sintering, it is more accurate to use the specific gravity method.
- Are all the experiments dried every 1 layer? Or is it dried after printing 5 layers? A clear expression is needed to this. Originally, the drying process should be performed for each single layer.
- In Figure 7, it is desirable to express the relative density or to indicate the density of the original material together.
- When indicating the test result, the number of repeated tests and the standard value of the material should be presented together. (density, bending strength at fracture and etc.) In addition, whether each experiment is a value before or after sintering should be clearly indicated.
- Are the microscopy images in Section 3.2.3 after sintering? The image is too blurry. It seems effective to add SEM pictures rather than optical microscopy pictures. It might also be that the material was divided into two layers due to sedimentation occurring in the coater.
- Are all experiments directly sintered after drying the slurry without binder jetting? Is there any collapse of the structure during sintering?
- If the result of Section 3.2.3 was after sintering, it is quite weird that there was no vertical shrinkage.
Author Response
Dear reviewer,
Thank you for taking the time and effort to review our manuscript. We sincerely appreciate your comments and hope we addressed all your concerns in the revised version as expected. Please find attached a list of changes related to your comments.
We integrated the changes to the revised manuscript in editing mode. In chapter 5 (conclusions) you will find further clarifications inspired by the other reviewers.
Thank you for your review that helped us to improve our paper.
Best regards,
Patricia Erhard

Reviewer 2 Report
The value of n in Fig. 6-Fig. 9 are not specified. The author needs to define the n value in the obvious place.
In line 414, the picture is missing from the manuscript. The author must add photographs and check for such errors.
In conclusion, the author writes that “The study focused on the characterization of specimens fabricated at different parameter sets related to drying, by altering the drying periods and intensities as well as the layer heights”. The author needs to express that the details about the relationship of different parameter and specimens characterization.
The manuscript uses density, 4-point bending strength and surface roughness to descript the physical properties of the specimens. Is there an optimal performance of the sample or an index which evaluates the overall performance of the sample? The author needs to explain it.
Author Response
Dear reviewer,
Thank you for taking the time and effort to review our manuscript. We sincerely appreciate your comments and hope we addressed all your concerns in the revised version as expected. Please find attached a list of changes related to your comments.
We integrated the changes inspired by all reviewers to the revised manuscript in editing mode. The addition of a scheme related to the drying theory (Fig. 3) is a noticeable difference to the original version. You will find some more small changes (eg. horizontal lines in Fig. 7 and 8) suggested by other reviewers.
Thank you for your review that helped us to improve our paper.
Best regards,
Patricia Erhard

Round 2
Reviewer 1 Report
I think appropriate corrections have been made.
Reviewer 2 Report
The manuscript has innovative and logical and is recommended for publication.